# Machine learning models based on remote and proximal sensing as potential methods for in-season biomass yields prediction in commercial sorghum fields

**Ephrem Habyarimana** [1]*, **Faheem S. Baloch** [2]

**1** CREA Research Center for Cereal and Industrial Crops, Bologna, Italy, **2** Faculty of Agricultural Sciences and Technologies, Sivas University of Science and Technology, Sivas, Turkey

* ephrem.habyarimana@crea.gov.it

**Data Availability Statement:** All relevant data are within the paper and its Supporting Information files.

## Abstract

Crop yield monitoring demonstrated the potential to improve agricultural productivity through improved crop breeding, farm management and commodity planning. Remote and proximal sensing offer the possibility to cut crop monitoring costs traditionally associated with surveys and censuses. Fraction of absorbed photosynthetically active radiation (fAPAR), chlorophyll concentration (CI) and normalized difference vegetation (NDVI) indices were used in crop monitoring, but their comparative performances in sorghum monitoring is lacking. This work aimed therefore at closing this gap by evaluating the performance of machine learning modelling of in-season sorghum biomass yields based on Sentinel-2-derived fAPAR and simpler high-throughput optical handheld meters-derived NDVI and CI calculated from sorghum plants reflectance. Bayesian ridge regression showed good cross-validated performance, and high reliability ($R^2$ = 35%) and low bias (mean absolute prediction error, MAPE = 0.4%) during the validation step. Hand-held optical meter-derived CI and Sentinel-2-derived fAPAR showed comparable effects on machine learning performance, but CI outperformed NDVI and was therefore considered as a good alternative to Sentinel-2's fAPAR. The best times to sample the vegetation indices were the months of June (second half) and July. The results obtained in this work will serve several purposes including improvements in plant breeding, farming management and sorghum biomass yield forecasting at extension services and policy making levels.

## Introduction

Under the climate change scenarios, the rapid increase of World population and industrial development are expected to give rise to increased carbon dioxide concentration in the Earth's biosphere, while environments are predicted to be warmer and dryer, all of which will favor the cultivation of crops with a C4 photosynthetic pathway over C3 crops [1–3]. Humans will therefore rely heavily on C4 crops like sorghum (*Sorghum bicolor* (L.) Moench). This crop is a

**Funding:** This research was funded by the European Union, grant number 732064 (H2020-ICT-2016-1-innovation action through the project Data-driven Bioeconomy (www.databio.eu), the Ministero delle Politiche Agricole, Alimentari, Forestali e del Turismo (Rome, Italy) through the project Risorse GeneticheVegetali (RGV/FAO) 2017–2019, and the The project SYSTEMIC "an integrated approach to the challenge of sustainable food systems: adaptive and mitigatory strategies to address climate change and malnutrition", Knowledge hub on Nutrition and Food Security, has received funding from national research funding parties in Belgium (FWO), France (INRA), Germany (BLE), Italy (MIPAAF), Latvia (IZM), Norway (RCN), Portugal (FCT), and Spain (AEI) in a joint action of JPI HDHL, JPI-OCEANS and FACCE-JPI launched in 2019 under the ERA-NET ERA-HDHL (n° 696295). The funders had no role in study design, data collection and analysis, decision to publish, or preparation of the manuscript".

**Competing interests:** The authors have declared that no competing interests exist.

staple cereal across countries under lower latitudes, but it is resurging under higher latitudes for feed, biofuel, the manufacture of specialty health-promoting foods rich in antioxidants, and for use as a substitute for traditional grains diet and meeting gluten-free needs [4–8]. Biomass sorghums of biofuel production interest were used in this work and were described in previous studies [4, 5]. They included dual purpose (showing high grain and biomass yields), forage, sweet, and biomass *per se* sorghum types [7, 8].

As sorghum is becoming increasingly used worldwide as a food and biofuel-dedicated biomass crop, its cultivation and yields will have to be closely monitored and forecast for efficient management locally and globally. The aim of the yield forecasting is to provide scientifically sound and independent estimates of crops' yield, as precise and as early as possible within the cropping season. Crop yield forecast supports farming operations through better planning of harvest, machineries, and logistics, and helps avoid food and commodity crises [4, 5, 9–12]. Crop yield forecasting can be achieved using three methods: field surveys, remote and proximal sensing models and crop simulation models [4, 5, 13]. Traditionally, yield forecasting was carried out by experts through the evaluation of crop status (e.g., tiller number, spikelet number, spikelet fertility percentage, percentage of biotic adversity damage, percentage of weeds infestation) throughout the crop growing season, and using these observations and measurements to forecast yields using regression methods, or relying on the experts knowledge [13]. This approach was nonetheless costly and error-laden [14]. Crop simulation models approximate the real world [15], and rely on computer simulations of crop growth, development and yield, solving mathematical equations with as covariates the soil conditions, weather and management practices [16].

In this work, proximal and remote sensing technologies were comparatively evaluated for sorghum biomass yield prediction purposes in commercial farmers' fields. Modern crop monitoring relying on remote and proximal sensing technologies resulted in a superior solution compared to traditional methods [14, 17–22]. Moreover, the use of remote and proximal sensing models to forecast within-season crop yields can outperform or be comparable to complex physiological crop simulation models [4]. The sensor-based monitoring relies upon differential reflectance of light by plants [23] which generally absorb the portion of light in the wavelength range of 400–700 nm (i.e., in the blue 440–510 nm, and red 630–685 nm wavelengths), and reflect light in the green and near infrared portions of the light spectrum. Crop monitoring technologies have been used to exploit this phenomenon, including satellites and hand-held sensors measuring light in narrow wavebands or wavelength intervals. Plant reflectance measurements have been successfully used in several modelling instances including the quantification of canopy vigor [24–26], nutrient and soil moisture stresses [27, 28], and to predict yields [4, 5].

Several statistical models implemented for yields forecasting purposes were amply described in previous studies [29–33]. Tucker et al. [30] used statistical models to uncover the relationships between yield and normalized difference vegetation index (NDVI), while [34] used reflectance data sampled at the grain-filling growth stage to forecast yields in rice. Benedetti and Rossini [35] used the Advanced Very High Resolution Radiometer satellite (AVHRR)-derived NDVI data at the grain-filling growth stage to monitor and forecast wheat yields in Italy, solving a simple linear regression model; the validation of this model against official data resulted in good correlations between NDVI and crop performance. In their study, Kogan et al. [20] came across a comparative performance between empirical NDVI and meteorological regression-based models and CGMS (Crop Growth Monitoring System) biophysical model in forecasting winter wheat yield at oblast level in Ukraine 2–3 months prior to harvest. Habyarimana et al. [4, 5] developed machine learning prediction models for sorghum biomass yields in Italy using the fraction of absorbed photosynthetically active radiation

(fAPAR) derived from Sentinel-2 constellations and came across medium to high accuracy in terms of models performance. Sentinell-2 crop and biophysical parameters can be processed on demand through the Sentinel-2 data service platform (https://s2.boku.eodc.eu) [36, 37] which provides atmospherically corrected images and biophysical variables including fAPAR, for any land surface on Earth.

Sensor technologies can now collect information at different spatial and spectral (wavelength) resolutions. Handheld spectroradiometers collect reflectance data by wavelength, with wavebands as precise as 1 nm [23], and they are typically designed for investigations requiring discrimination between a high number of observations or as research tools to develop and test methods for specialized sensors. A generation of handy specialized handheld sensors such as FieldScout CM 1000 (Spectrum Technologies Inc, 360 Thayer Court, Aurora, IL 60,504) used in this work, was produced building upon complex spectroradiometers, and their use simplified crop monitoring [23, 38]. FieldScout CM 1000 sensors use reflectance in the red (NDVI meter), red-edge (690–730 nm) (chlorophyll meter) and near-infrared (NIR) regions to derive plant chlorophyll concentrations. The red edge describes the steeply sloped region of the vegetation reflectance curve between 690 nm and 730 nm that is caused by the transition from chlorophyll absorption and near-infrared leaf scattering [39]. Red-edge spectral reflectance is therefore relatively easily affected (deeply controlled) by chlorophyll content changes compared to the red spectral reflectance. When used over closed canopy, handheld measurements are comparable to multispectral imaging technologies that were developed to measure reflectance in the red and NIR regions to estimate the amount of photosynthates at pixel level [23]. Normalized difference vegetation index (NDVI), leaf area index (LAI), chlorophyll concentration index (CI), and fraction of absorbed photosynthetically active radiation (fAPAR) are among parameters most frequently used as proxies for biomass yields in several crops [23, 38, 40]. The fraction of absorbed photosynthetically active radiation is defined as the fraction of radiation absorbed by the green vegetation elements in the 400 to 700 nm spectral domain [41], while NDVI and CI sample PAR in the red (at 660 nm) and red-edge (at 700 nm), respectively [23, 39, 42].

Most of the research works on remote and proximal sensing for sorghum monitoring and yields prediction were focused on single biophysical and/or crop parameters, but the comparative performance of machine learning models derived from the factorial combination of CI, NDVI, and fAPAR fitting the same dataset, is lacking [19, 29, 30, 43–45]. This work was therefore undertaken to address this gap. For instance, Johnson [19] investigated the correlation of several Moderate Resolution Imaging Spectroradiometer (MODIS) composited imagery products with crop yields in ten global agricultural commodities (barley, canola, corn, cotton, potatoes, rice, sorghum, soybeans, sugar beets, and wheat) using NDVI, fAPAR, LAI, Gross Primary Production (GPP), daytime Land Surface Temperature (DLST) and nighttime LST (NLST). Habyarimana et al. [4, 5] used Sentinel-2a- and Sentinel-2b-derived fAPAR to predict sorghum biomass yields using supervised parametric and nonparametric machine learning approaches. Shafian et al. [46] evaluated the performance of an Unmanned Aerial Systems-based remote sensing system for quantification of crop growth parameters of sorghum LAI, fractional vegetation cover (fc) and yield. In their work to assess the potential use of portable SPAD-502 and the FieldScout CM 1000 NDVI meters in diagnosing the nutritional status of plants, Afonso et al. [47] concluded that although both devices showed good reproducibility, the FieldScout NDVI readings showed a marked saturation curve with the leaf nitrogen concentration, and was thus considered not suited for reliable use as nitrogen nutritional status index. Using RapidEye images and in situ data from alfa-alfa, barley, maize and winter wheat, Xie et al. [39] proposed three improved indices combining reflectance both in the red and red-edge spectral regions into the NDVI, the modified simple ratio index (MSR), and the green

chlorophyll index formula for LAI retrieval. Interestingly, the authors demonstrated that the reflectance in the red-edge spectral region was sensitive to chlorophyll concentrations and enabled to account for their high variability across crops and phenological states.

The specific objective of this study was to comparatively evaluate the importance of Sentinel-2-derived fAPAR and handheld high-throughput proximal optical reflectance meters sensors for the prediction of biomass yields using supervised parametric and nonparametric machine learning models. This work represents therefore an attempt to push current methods towards opening up new avenues of research including the use of simpler but robust handheld sensors. The research questions addressed in this work were: (1) can information derived from hand-held optical high-throughput chlorophyll meters represent an alternative to Sentinel-2 imagery in terms of machine learning predictive analytics of biomass sorghum yields? (2) Which months or parts thereof best contribute useful information for predicting biomass yields in commercial sorghum fields? The first question is of particular relevance in the Mediterranean region where coexist small and big farms, individual farmers and farming cooperatives with no or few infra-structures to process satellite imageries. The rationale is, while it is an obliged strategy for the extension services to use satellite imageries to monitor crops at a regional level and beyond, it can nonetheless be handy for farming cooperatives to use a simple but accurate and high-throughput hand-held sensor to monitor fields at a relatively smaller scale. Some of the most attractive features of the optical chlorophyll meters used in this work is that they are simple, easy to use in high-throughput phenomics measurements, are more affordable, more farmer friendly because they are easy to operate, requiring little technical skills and facilities and not entirely dependent on time and resource consuming calibration exercise [42].

## Materials and methods

### Open field trials

A total of twenty-three trials were established in this work, of which nineteen and four, respectively, were evaluated in 2018 and 2019. In 2018, the trials were located in Conselice, Miran-dola, and Anzola dell'Emilia, while in 2019 the experimental sites were located in Conselice (Fig 1 and Table 1); the geographic coordinates were included in Table 1. The trials conducted in 2018 were described in our previous works [4, 5]. The fields in Anzola are part of the experimental station of the Council for Agricultural Research and Economics (CREA), while those in Mirandola and Conselice belonged to two private commercial cooperatives. Formal legally binding contracts were signed between CREA Consiglio per la ricerca in agricoltura e l'analisi dell'economia agraria and the cooperatives, in order to regulate field trials management and authorize CREA to access the fields and publish related information. Sizeable experimental fields were used [6] in this study in order to not only increase the precision of the data collected, but also to use the experiments as demonstration sites for technology transfer into the production environments. The fields areas ranged from 0.06 to 45.00 ha, with a median and mean of 3.72 ha and 7.60 ha, respectively. The trials were sown with biomass sorghums; twenty trials were planted with a single genotype of *Sorghum bicolor* (annual), while the other three trials (15R18, 16R18 and 17R18) consisted of populations regrown from overwintered rhi-zomes of the 2015, 2016 and 2017 trials, respectively. The 15R18, 16R18 and 17R18 trials repre-sented three spatio-temporal replications of a diversity panel of advanced perennial interploid biomass recombinant inbred lines deriving from *S. bicolor* × *S. halepense* (SB × SH) crosses. The 15R, 16R, and 17R trials were sown in 2015, 2016, and 2017, respectively, meaning that the respective populations (15R18, 16R18 and 17R18) underwent three, two and one overwin-tering cycles, respectively. Crops were managed according to the guidelines of the extension services in the region of Emilia Romagna [6]. Trials were planted at a density of 26 (0.75 m

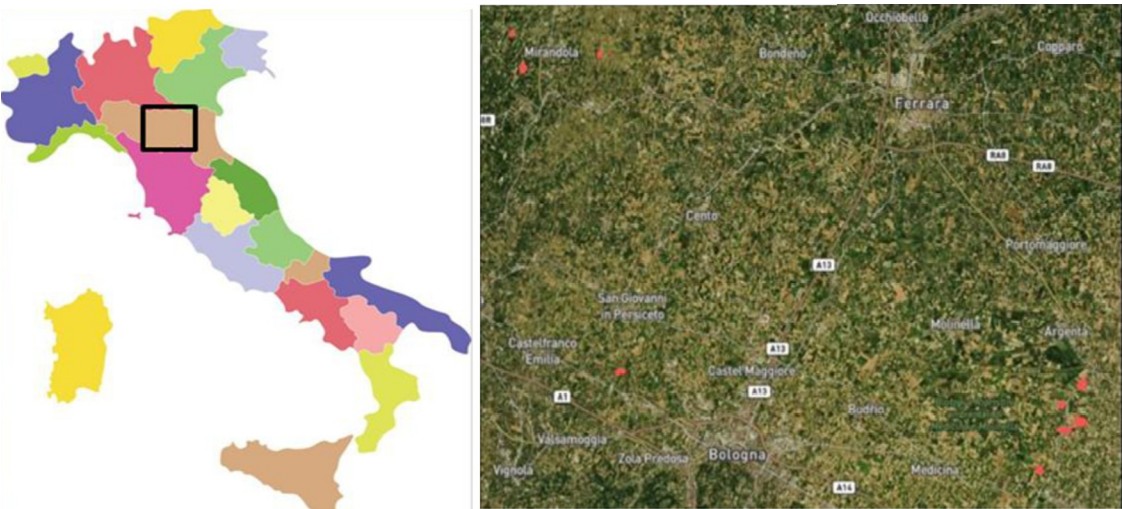

**Fig 1. Italian map (left): The rectangle inset indicates the geographical locations of the trials (red dots) set-up in 2018 and 2019 (right).** Red dots: upper-left, lower-left, and lower-right, respectively, trials location in Mirandola, Anzola dell'Emilia, and Conselice. Refer to the text for the description of the field trials. Reprinted from Habyarimana et al. [4] under a CC BY 4.0 license (http://creativecommons.org/licenses/by/4.0/), original copyright 2019 by the authors".

spacing between rows; 0.052 m spacing of hills within row) plants per square meter for most (20) trials, and 13 (0.75 m spacing between rows; 0.10 m spacing of hills within row) plants per square meter for 15R18, 16R18, and 17R18.

**Table 1. Experimental sites and dimensions, sorghum varieties sown, and aboveground biomass yields.**

| Trial Name | Variety | Area (ha) | Yield (t/ha) | Year | Location | Latitude N | Longitude E |
|---|---|---|---|---|---|---|---|
| Val Serrata | Harmattan | 44.87 | 9.69 | 2018 | Conselice | 44°31'32.12" | 11°51'18.82" |
| Magnana | P845F | 32.05 | 11.43 | 2018 | Conselice | 44°33'44.81" | 11°51'25.37" |
| Botte_1_2018 | Harmattan | 9 | 14.13 | 2018 | Conselice | 44°28'42.12" | 11°47'36.84" |
| Gamberina_3 | Aralba | 7.86 | 9.67 | 2018 | Conselice | 44°32'30.06" | 11°49'37.05" |
| Ca_bianca | P845F | 3.72 | 11.11 | 2018 | Conselice | 44°31'04.04" | 11°49'33.00" |
| Saracca_5 | Harmattan | 6.5 | 10.52 | 2018 | Conselice | 44°31'04.18" | 11°50'12.63" |
| Anzola-T4_18 | Trudan | 0.7 | 10.02 | 2018 | Anzola | 44°34'32.32" | 11°10'12.17" |
| Anzola-T3_18 | Tarzan | 0.71 | 10.47 | 2018 | Anzola | 44°34'31.12" | 11°10'15.35" |
| Anzola-T1_18 | Bulldozer | 0.74 | 6.32 | 2018 | Anzola | 44°34'28.76" | 11°10'21.73" |
| Anzola-T5_18 | Harmattan | 0.7 | 3.97 | 2018 | Anzola | 44°34'33.25" | 11°10'08.41" |
| Anzola-T2_18 | Hannibal | 0.71 | 10.84 | 2018 | Anzola | 44°34'29.97" | 11°10'18.42" |
| CAMA_grande_18 | Bulldozer | 4.4 | 14.94 | 2018 | Mirandola | 44°52'12.28" | 11°01'26.20" |
| Lidia_18 | Bulldozer | 3 | 7.19 | 2018 | Mirandola | 44°53'06.96" | 11°08'17.15" |
| 16R18 | Perennial | 0.15 | 14.35 | 2018 | Anzola | 44°34'26.92" | 11°09'56.75" |
| CAMA_picolo_18 | Bulldozer | 4 | 15.35 | 2018 | Mirandola | 44°52'13.01" | 11°01'19.43" |
| 15R18 | Perennial | 0.06 | 10.73 | 2018 | Anzola | 44°34'31.26" | 11°10'01.27" |
| 17R18 | Perennial | 0.15 | 17.47 | 2018 | Anzola | 44°34'24.40" | 11°09'52.78" |
| Raimondo_18 | Bulldozer | 2.02 | 10.48 | 2018 | Mirandola | 44°54'18.33" | 11°00'26.00" |
| Villa_Verdetta_18 | Bulldozer | 1 | 6.25 | 2018 | Mirandola | 44°52'21.94" | 11°01'23.02" |
| Bassa fornace 1 | Swing | 9.41 | 6.05 | 2019 | Conselice | 44°31'38.96" | 11°49'3.55" |
| Cassa Cornacchiona | Aralba | 22.19 | 16.78 | 2019 | Conselice | 44°32'41.06" | 11°49'53.56" |
| Piana 7 | Harmattan | 14.52 | 12.24 | 2019 | Conselice | 44°33'59.47" | 11°51'7.27" |
| Punta FS Punta Botte | Tresor | 6.39 | 8.76 | 2019 | Conselice | 44°32'55.53" | 11°50'27.00" |

## Trials biomass measurements

Trials were mechanically harvested from the end of August to late November using forage chopper or swathing the biomass material into windrows and baling it in square or round bales. Chopped and baled biomasses were weighed, respectively, at the time of harvesting and at the time of transportation to the bioreactor. Chopped material was supplied to private biogas bioreactor company, while baled material was supplied to a local combustion bioreactor company. The process of biomass sampling and determination dry biomass yields was carried out as suggested by Habyarimana et al. [4, 5]. Briefly, a composite biomass sample of approximately 0.5–1 kg was collected for each genotype individually; the sample fresh weight was immediately measured, while the dry weight was determined after drying the sample at 80˚C in a forced air oven until the weight was constant. The dry mass fraction of fresh material (DMC) was calculated as the ratio of dry/fresh sample weights, and the dry biomass yield was derived multiplying DMC by the fresh weight of the plot or field's harvest, expressed in t/ha.

## Proximal meters reflectance sampling

Normalized Difference Vegetation Index (NDVI) and leaf chlorophyll concentration index (CI) were measured from day of year (DOY) 164 towards the end of the fast growth stage to DOY 264 using FieldScout CM 1000 NDVI and FieldScout CM 1000 chlorophyll meters (Spectrum Technologies Inc, 360 Thayer Court, Aurora, IL 60,504) on a weekly (avoiding cloudy days) basis on days when Sentinel-2A and Sentinel-2B satellites coverage was 100% in the region where the trials were established. The FieldScout CM 1000 NDVI and FieldScout CM 1000 chlorophyll meters were designed to measure chlorophyll concentration in the leaves, and they were used according the manufacturer's instructions. They utilize laser directed "point and shoot" technology to rapidly measure light transmittance in the red (at 660 nm) for NDVI meter, red-edge (at 700 nm) for chlorophyll meter) and near-infrared (840 nm) spectral bands. On a data collection day, 250 optical chlorophyll and NDVI measurements were taken from the canopy in each field with the FieldScout CM 1000 meters, and the average was calculated to represent the reading average of each field on that day. The meters were operated manually by holding them approximately between 40 and 150 cm from the sorghum plants canopy at a 45–90˚ vertical or horizontal angles depending of the plant height at the time of measurements. The same operator used the same meter across seasons, and all meters sampled the same area in the field; single measurements were taken in the third row, skipping two rows from the border of the field, by gently walking and shooting lasers at the canopy or the upper parts of the plants in a row, at approximately two-second intervals of 0.60 m distance, each, ultimately covering 150 m of sampled row length in each field. As per the manufacturer's specifications, the FieldScout meters are equally accurate within a canonical field of view ranging from 28.4 to 183 cm. The instruments use two diode lasers to define the target area to be sampled as the trigger is pressed: at a distance of 28.4 cm, the field of view is 1.10 cm in diameter, while at a distance of 183 cm, the field of view increases to 18.8 cm in diameter [48]. The normalized difference vegetation index (NDVI) is defined as in the following equation $(\rho_{NIR} - \rho_{RED})/(\rho_{NIR} + \rho_{RED})$ where $\rho$ stands for the reflectance values at the wavelength of interest. The NDVI and chlorophyll concentration indices were measured on a scale of -1 to 1 and 0 to 999, respectively. The laser guide lights were used to aim the meter at target row sections; the laser was focused at different heights within the upper canopy of the plants, avoiding older leaves. The meter readings were taken between 10 a.m. and 2 p.m. as suggested in literature [40, 42], with the sun to the back of the reader without shading the ambient light receiver. Reflectance readings were stored in the meter and uploaded in the laptop at the end of each

sampling day, for downstream analyses. The NDVI and CI values used in the predictive modelling presented in this work are averages on a fortnight basis.

## Sentinel-2 fAPAR data acquisition

The Copernicus Sentinel-2 optical satellite imageries were used in this work as described previously [4, 5]. The Sentinel-2 mission is made up of a constellation of two polar-orbiting satellites placed in the same sun-synchronous orbit at an altitude of 786 km, and phased at 180˚ to each other to optimize coverage and revisit times. The mission displays a wide swath width (290 km) and high revisit time of ten days at the equator when one satellite is considered, and five days with 2 satellites, resulting in two to three days under mid-latitudes. The coverage limits of the Copernicus Sentinel-2 mission ranges from latitudes 56˚ south to 84˚ north. The multispectral instrument (MSI) of Sentinel-2 measures the radiance reflected by Earth in thirteen spectral bands from VNIR (visible and near-infrared) to SWIR (short-wave infrared), of which four, six, and three bands with a spatial resolution (surface area measured on the ground, represented by one pixel) of, respectively, 10 m (blue, green, red, and NIR), 20 m (3 red edge bands, 1 narrow NIR, 2 SWIR), and 60 m (a coastal aerosol, water vapor, and cirrus band). After Sentinel data are acquired, they are sent to Earth for subsequent processing, archiving, and dissemination through ESA's Copernicus Space Component (CSC) Ground Segment. Images projected from Sentinel-2 are converted to individual tiles of 100 km$^2$ stored in respective files of approximately 500 megabytes. In this work, two tiles of interest i.e., 32TQQ (including Conselice) and 32TPQ (including Anzola and Mirandola) were downloaded into WatchITgrow (VITO, Vlaamse Instelling voor Technologisch Onderzoek N.V., MOL, Belgium) for downstream processing The downloaded satellite images were processed using iCOR [49] for atmospheric correction and Sen2COR v2.5.5 for cloud and shadow detection (ESA-STEP, ESA, Paris, France). The fAPAR parameter along with other biophysical parameters e.g., fCover and leaf area index (LAI) were computed from the top of canopy normalized reflectances using the BV-NET (tool for surface and vegetation variables mapping) method as suggested in Weiss and Baret [50]. In the BV-NET, the neural networks algorithms are trained on a synthetic dataset of around 50,000 simulations using the PROSAIL (PROSPECT and SAIL radiative transfer models) model [51]; the algorithms are trained using green, red and near infrared bands with 10-meter spatial resolution. Sen2Cor and BV-NET are publicly available through ESA's SNAP (Sentinel Application Platform, ESA, Paris, France) toolbox. Instantaneous fAPAR values at the time Sentinel-2 overpasses are computed directly from the radiative transfer model in the green parts of the canopy, and have a spatial and temporal resolutions of 10 m and 5 days, with up to two to three days areas with satellite overlapping overpasses.

Before crop monitoring with satellite-derived fAPAR, the trials were geolocalized (Fig 1) using Field GPS (global positioning system) application for iPhone, and the final field boundary corrected using Google Earth. The fAPAR values of all pixels within individual fields were averaged, accounting for a ten-meter boarder (1 pixel) to avoid signals from neighboring fields and nontarget objects. Whittaker smoothing filter was implemented on the fAPAR curve [52, 53] in order to correct for artifacts such as abnormally low fAPAR values that can arise from undetected clouds, shadows or haze, and to interpolate fAPAR between acquisition days. In the predictive analytics presented in this work, fAPAR values were used that corresponded to fortnightly-spaced single days of year and/or within-fortnights averages. Average values were considered in order to accommodate handheld sensor data collection that was at times interrupted by rainfall and did not coincide with 100% satellite coverage of the trial locations. The

preference was given to the use of fortnightly reflectance data because major morphophysiological changes in crops occur fortnightly [54], and this planning accommodated farmers and scientists schedules.

## Machine learning modelling and statistical inferences

Eleven models were assessed in this study to predict sorghum biomass yields using 23 instances and 24 features (D120_F, D135_F, D150_F, D165_F, D180_F, D195_F, D210_F, D225_F, D240_F, D164_176F, D184_199F, D202_215F, D225_239F, 249_264F, D164_176C, D184_199C, D202_215C, D225_239C, 249_264C, D164_176N, D184_199N, D202_215N, D225_239N, 249_264N) as shown in Tables 1 and 2 and S1. The models included Linear Model (LM), Neural Network (nnet), eXtreme Gradient Boosting (xgbTree), Bayesian Ridge Regression (bridge), and Random Forest (rf) fitting all regressors, on the one hand, and on the other hand, bridge x fAPAR, bridge x NDVI, bridge x CI, bridge x NDVI x CHL, bridge x fAPAR x NDVI, bridge x fAPAR x CI representing models in which Bayesian ridge regression was implemented in factorial combinations of the sensor data types. Linear model and Bayesian ridge regression are parametric models; in the linear model, the unknown parameters are estimated using the least-squares estimation, while the Bayesian ridge regression is a version of ridge regression in which the features' effects are normally distributed (i.e., Gaussian prior),

**Table 2. Features used in the predictive models and relative descriptive statistics.**

| Features[†] | Minimum | First Quartile | Median | Mean | Third Quartile | Maximum |
|---|---|---|---|---|---|---|
| D120_F | 0.0918 | 0.1608 | 0.1837 | 0.2017 | 0.2468 | 0.3143 |
| D135_F | 0.0601 | 0.1403 | 0.2028 | 0.1952 | 0.2559 | 0.3466 |
| D150_F | 0.0957 | 0.2014 | 0.2475 | 0.2727 | 0.3248 | 0.5300 |
| D165_F | 0.1532 | 0.3740 | 0.4848 | 0.4570 | 0.5632 | 0.6743 |
| D180_F | 0.2687 | 0.6660 | 0.7062 | 0.6652 | 0.7514 | 0.8074 |
| D195_F | 0.3011 | 0.7425 | 0.7909 | 0.7611 | 0.8241 | 0.8958 |
| D210_F | 0.2523 | 0.7600 | 0.8046 | 0.7813 | 0.8416 | 0.9076 |
| D225_F | 0.2420 | 0.6627 | 0.7345 | 0.7230 | 0.8254 | 0.9045 |
| D240_F | 0.0905 | 0.4565 | 0.6333 | 0.5815 | 0.7802 | 0.8823 |
| D164_176F | 0.1709 | 0.4930 | 0.5883 | 0.5397 | 0.6184 | 0.7393 |
| D184_199F | 0.2916 | 0.7195 | 0.7764 | 0.7358 | 0.8111 | 0.8731 |
| D202_215F | 0.2568 | 0.7656 | 0.8063 | 0.7825 | 0.8302 | 0.9017 |
| D225_239F | 0.2325 | 0.5378 | 0.6973 | 0.6690 | 0.8137 | 0.9228 |
| D249_264F | 0.0510 | 0.1339 | 0.3114 | 0.3802 | 0.6088 | 0.8717 |
| D164_176C | 201.7 | 358.4 | 422.2 | 423.5 | 478.9 | 647.4 |
| D184_199C | 381.2 | 534.3 | 546.9 | 558.2 | 576.1 | 724 |
| D202_215C | 367.4 | 438.5 | 519.9 | 547.4 | 621.7 | 864.4 |
| D225_239C | 204.8 | 395.4 | 455.6 | 484.4 | 533.8 | 875.1 |
| D249_264C | 262.0 | 439.2 | 439.2 | 444.1 | 439.2 | 716.1 |
| D164_176N | 0.4650 | 0.7931 | 0.813 | 0.7891 | 0.8403 | 0.8900 |
| D184_199N | 0.5567 | 0.8416 | 0.8600 | 0.8335 | 0.8679 | 0.8862 |
| D202_215N | 0.8132 | 0.8346 | 0.8507 | 0.8526 | 0.8758 | 0.8962 |
| D225_239N | 0.6247 | 0.7575 | 0.8177 | 0.7983 | 0.8569 | 0.8969 |
| D249_264N | 0.7268 | 0.8118 | 0.8118 | 0.8125 | 0.8118 | 0.8943 |

[†] D, C, F, N, respectively, day of year, chlorophyll concentration index, fraction of absorbed photosynthetically active radiation, normalized difference vegetation index. Numbers or numbers linked with underscore following the letter "D" are single day of year or intervals of days of year. Refer to the text for the description of these variables.

have identical variance and are shrunk toward zero [55]. Random Forest, Neural Network, and Extreme Gradient Boosting are nonparametric methods as the number of parameters in the model are not fixed [56, 57], but changes depending upon the data used during the training step; the number of parameters usually increases as the training dataset size increases [13]. Bayesian Ridge Regression was used to evaluate the relative importance of the sensor data types as it combined high performance, low complexity which avoid overfitting the model to the data [58], and short running time (e.g., 10 seconds *vs.* 15 minutes for xgbTree) on a laptop with 16 GB of RAM. The simple linear model was used as a benchmark to measure the relative performance of the models implemented. The other models (xgbTree, nnet, rf) evaluated in this work were selected based on the robustness they displayed in previous studies [4, 5, 59, 60].

The solutions to the above research questions addressed in this work were evaluated by solving the below linear model for *n* trials (*i* = 1,. . . .,*n*) and *p* prediction times across sensor data types (*j* = 1,. . . .,*p*). This model is represented by

$$y_i = \mu + \sum_{j=1}^{p} x_{ij}\beta_j + e_i \tag{1}$$

where $\mu$ is the overall mean, $y_i$ the phenotypic observation from trial *i*, $e_i$ the residual comprising all other nongenetic and environmental factors, $x_{ij}$ the sampling time x sensor data type covariates, and $\beta_j$ the effect of the $j^{ith}$ sampling time x sensor data type covariate on $y_i$ [61]. The scope here is to predict biomass yields over entire commercial fields and not to identify and/or predict within-field yield variability.

Statistical analyses were carried out using R software [62] and the predictive models fitted using R packages implementing the predictive methods of interest. To avoid overfitting, the "one standard error" rule of Breiman et al. [57] was used, and the methods' built-in features were invoked to automatically select features, tune hyperparameters to the data set, and select the best final model for the downstream validation step. All models implemented in this study were multicollinearity resistant except the linear model [59] which therefore required additional measures to avoid overfitting. In the linear model, in order to minimize the effect of collinearity, solve the issue of the number of predictor variables being greater than the sample size, estimate all parameters including the constant, and finally be able to fit the overall model during the training stage, we opted to reduce the number of predictors by using an algorithm to remove a subset of those features involved with the most high pairwise correlations such that the sample (training set) size is two more than the number of predictors (allowing for a residual degree of freedom), and all of the remaining pairwise Pearson correlation coefficients are below a 0.90 threshold [59]. In the process of data preparation, zero-variance features were removed and those remaining were centered and scaled in order to avoid features with zero or near-zero variance which can behave like second intercepts [59]. The dataset was randomly partitioned into training/calibration (70% of the entire dataset i.e., 16 observations) and testing/validation set (30% of the entire dataset i.e., 7 observations). It was necessary to have a testing set of at least six observations which is the minimum required size to run parametric statistical inferences [63]. The calibration set was used to run a cross-validation experiment to train and assess the models using a 10× repeated 5-random fold cross-validation iterations, generating a total of 50 estimates of accuracy and prediction biases. As suggested by Habyarimana et al. [4], a large number of repetitions can compensate for the high variance that can arise from low number of folds. Models were validated on the external testing set so that the model performance can be quantified on data that were not used during model calibration. During the calibration step, mean absolute error (MAE), root mean square error (RMSE), and the coefficient of determination ($R^2$) were used to evaluate the performance of the models.

During the validation step, the models were evaluated based on $R^2$, MAE, mean absolute percentage error (MAPE), and symmetric mean absolute percentage error (SMAPE). The MAPE allows to compare predictions of different dependent variables that were measured using different scales. The MAE is a measure of the average magnitude of the bias in the predicted values without accounting for their direction. In that sense, MAE gives unambiguous measure of the average bias (error) and can be considered as more appropriate than the Root Mean Square Error (RMSE). As for SMAPE, it averages the absolute percentage errors like in MAE, but the errors are calculated using a denominator representing the average of the forecast and observed values. The upper limit of SMAPE is 200%, meaning a 0 to 2 range, which simplifies the evaluation of accuracy, and as such, SMAPE is less sensitive to extreme values. In addition, SMAPE corrects for the computation asymmetry of the percentage error [64]. The distributions of the 50 MAE and $R^2$ estimates obtained from the cross-validation profiles were used to characterize and run inferences on the final models using boxplots and Student's t test [59]. The importance of the regressor variables (useful prediction times x data types) was determined using a 0 to 100 index, with 0 corresponding to no effect and 100 corresponding to the highest magnitude of the regressor's effect as suggested in literature [17, 65]. The accuracy or reliability of the models was defined as the Pearson correlation coefficient between the predicted and the observed biomass yield values in the validation set [66]. From the computed accuracy, the coefficient of determination can be straightforwardly derived in order to better compare, for each model, the proportion of the variance in the dependent variable that is predictable from the regressors.

For $n$ observations ($i = 1,2,\ldots.n$), the formulas of the model evaluation metrics used in this work are presented below:

$$MAE = \frac{1}{n}\sum\nolimits_{i=1}^{n} |y_i - \hat{y}_i| \tag{2}$$

$$MAPE = \frac{1}{n}\sum\nolimits_{i=1}^{n} \left|\frac{y_i - \hat{y}_i}{y_i}\right| \tag{3}$$

$$SMAPE = \frac{1}{n}\sum\nolimits_{i=1}^{n} \frac{2|y_i - \hat{y}_i|}{|y_i| + |\hat{y}_i|} \tag{4}$$

$$RMSE = \sqrt{\frac{1}{n}\sum\nolimits_{i=1}^{n} (y_i - \hat{y}_i)^2} \tag{5}$$

$$R^2 = 1 - \frac{\sum_{i=1}^{n} (y_i - \hat{y}_i)^2}{\sum_{i=1}^{n} (y_i - \bar{y}_i)^2} \tag{6}$$

where, $y_i$, $\hat{y}_i$, and $\bar{y}$ denote, respectively, an observation, its forecast and its mean taken over n, while "||" denotes absolute value.

## Results

### Descriptive statistics of the features used in modelling

The descriptive statistics of the reflectance data used in this study are presented in Table 2. Across cropping seasons and experimental sites, fAPAR data sampled on single days fortnightly spaced showed the highest values (0.76, 0.84, 0.80, 0.78, 0.91) of the first and third quartiles, median, mean and maximum on the day of year (DOY) 210 i.e., in the second half of

July. When within-fortnights fAPAR averages were used, the same trend was observed for the first and third quartiles, median, mean (0.77, 0.83, 0.81, 0.78, respectively), but the maximum (0.92) was recorded in the second half of August on DOY 225–239. Chlorophyll concentration index (CI) data showed highest values (534.30, 546.90, 558.20) of first quartile, median, and mean on DOY 184–199 (first half of July), while highest values (621.70, 875.10, respectively) of the third quartile and the maximum were recorded on DOY 202–215 (second half of July) and 225–239, respectively. Normalized difference vegetation index data displayed highest values (0.84, 0.86) of first quartile and median on DOY 184–199, while peaks for the mean and third quartile were observed on DOY 202–215 and those for the maximum occurred on DOY 225–239. The peaks for the minimum values were recorded in the first half of July on DOY 195 for single fortnights fAPAR and on DOY 184–199, each, for within-fortnights averages of fAPAR and CI, while such peaks were observed on DOY 225–239 for within-fortnights NDVI averages.

## Models performance using all features

During the cross-validation process, mean absolute error (MAE) was used to select the optimal model using the "one standard error rule" as suggested by Hastie et al. [67]. This allowed to produce smoother models than those directly estimated by automatic smoothing parameter selection methods. Tuning parameters were used in nonparametric models in order to fit the dataset. In Random Forest (RF), the final value used for the model was mtry (number of randomly selected predictors) = 2 (Fig 2). In Neural Network (NNET), the final values used for the model were size (number of hidden units) = 1 and decay (weight decay) = 0.1 (Fig 3). In the case of Extreme Gradient Boosting (xgbTree), the tuning parameters "gamma" (minimum loss function) and "min_child_weight" (minimum sum of instance weight) were held constant at values of 0 and 1, respectively, and the final values used for the model were nrounds (number of boosting iterations) = 50, max_depth (max tree depth) = 1, eta (shrinkage or learning rate) = 0.3, gamma (minimum loss function) = 0, colsample_by_tree (subsample ratio by column) = 0.6, min_child_weight = 1 and subsample (subsample percentage) = 0.75 (Fig 4).

The cross-validation performance range of LM, RF, BRR, NNET, and xgbTree models was, respectively, 4.76–1588.38, 0.66–5.74, 0.55–6.11, 6.24–13.90, 0.67-7-37 for MAE, 5.20–1894.30, 0.72–5.77, 0.59–6.62, 6.98–14.26, 0.73–8.18 for RMSE, and 0.0002–1, 0.00002–1, 0.0007–1, 0.0002–1, and 002–1 for $R^2$ (Fig 5). A comparative evaluation of model performance using MAE and the test of Student showed significant pairwise differences between models except BRR *vs*. xgbTree, RF *vs*. BRR, and RF *vs*. xgbTree (Fig 6). All the algorithms outperformed the linear model, Random Forest outperformed Neural Network but was comparable to Extreme

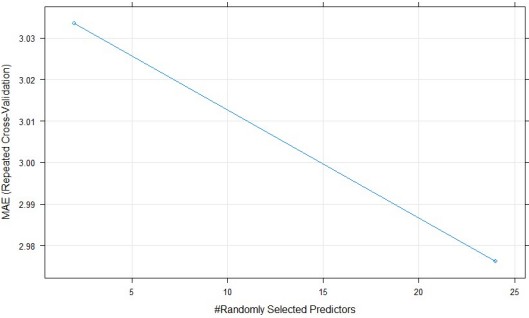

**Fig 2. Cross-validation profile and parameter tuning for Random Forest (RF) model.** MAE, #, respectively, mean absolute error, number.

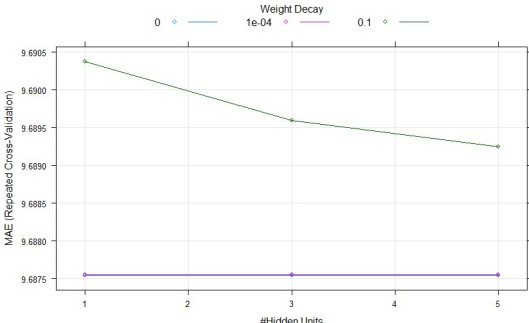

**Fig 3. Cross-validation profile and parameter tuning for Neural Network (NNET) model.** MAE, #, respectively, mean absolute error, number.

Gradient Boosting and Bayesian Ridge regression. Bayesian ridge regression outperformed neural network but was comparable to Extreme Gradient Boosting. When the calibrated models were used to predict the testing set sample that was not used during model training, all models but LM displayed MAPE < 1% (Table 3). Bayesian ridge regression showed the highest (35%) $R^2$ (coefficient of determination) value followed by Neural Network (20%); the remaining models displayed poor $R^2$ ranging from 1 to 6%. In the context of this work, the coefficient of determination represents the proportion of the variance in the testing set that is predictable from the model i.e., it measures of how well the testing set is replicated by the model, based on the proportion of total variation in the testing set explained by the model.

## Determination of the importance of sensors and vegetation indices sampling time using Bayesian ridge regression

Bayesian ridge regression combined high value of the coefficient of determination and good precision metrics such as MAE, MAPE, and SMAPE (Table 3) and was therefore used to evaluate the importance of the different combinations of sensors and vegetation indices sampling times used in this work (S1 Table and Figs 7 and 8). A factorial combination of the different sensor data using BRR produced seven models: FAN (fAPAR x NDVI), FAP (fAPAR), FAC (fAPAR x CI), NDCL (NDVI x CI), ALL (fAPAR x NDVI x CI), NDV (NDVI), and CHL (CI) (Fig 7). The cross-validation of these models produced the distribution of the evaluation metrics (MAE, RMSE, $R^2$) as presented in Fig 7. Briefly, MAE ranged from 0.33 (CHL) to 6.69 (FAN), RMSE from 0.34 (CHL) to 7.15 (FAN), and $R^2$ from 0.0002 (FAP) to 1. A comparative

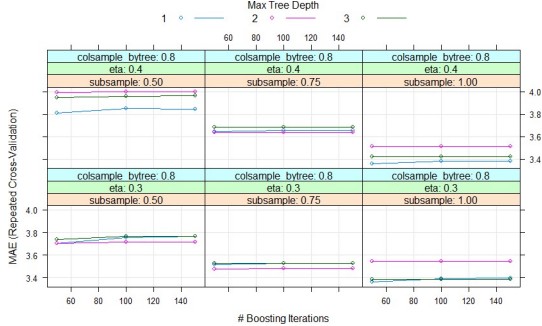

**Fig 4. Cross-validation profile and parameter tuning for Extreme Gradient Boosting (xgbTree) model.** MAE, #, respectively, mean absolute error, number.

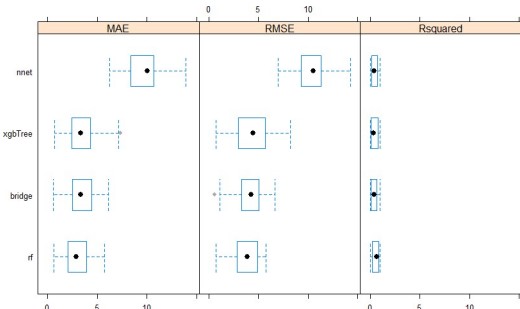

**Fig 5. Distribution (boxplot) of cross-validation performance metrics for the machine learning algorithms implemented.** MAE, RMSE, Rsquared, rf, xgbTree, nnet, bridge, respectively, mean absolute error, root mean square error, coefficient of determination, random forest, extreme gradient boosting, neural network, Bayesian ridge regression. The distribution for the linear model was not included in the graphic as it displayed higher bias values with wider dispersion making the scale illegible for more accurate algorithms. Refer to text for the description of the distribution and the algorithms.

evaluation of the performance of these models using MAE and the test of Student produced 21 pairwise comparisons (Fig 8) of which only eight resulted in significant differences ($P < 0.05$): NDV, NDCL, FAC, and ALL outperformed FAN, while CHL outperformed NDV, NDCL, FAN, and ALL, but was comparable to FAP.

The analysis of the importance of the vegetation indices sampling time across sensor data types was carried out using a 0 to 100 index, with 0 corresponding to no effect, while 100 corresponds to the highest magnitude of the feature's importance. The average reflectance data sampled on days-of-year 202–215 (D202_215C) and 164–176 (D164_176C) using chlorophyll concentration index resulted in the two most important regressors using Bayesian Ridge Regression (Fig 9). These two sampling times (DOY 164–176 and DOY 202–215) correspond to the first half of June and July, respectively. They were followed, in decreasing order of importance, by DOYs 249–264 in early to mid-September using NDVI, 120 in end April using fAPAR, 164–176 using NDVI, 184–199 in early to mid-July using CI, 180 in end June and 184–199 using fAPAR, 225–239 in second half of August using NDVI, and 150 in end May using fAPAR. When BRR was implemented on chlorophyll concentration index data only, reflectance sampled on DOYs 202–215 and 164–176 resulted in the two most important regressors with importance index of 100,

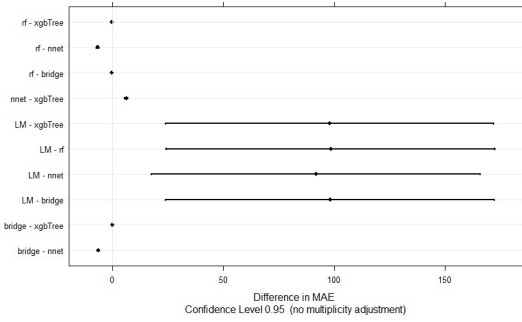

**Fig 6. Pairwise comparison of the machine learning algorithms using Student's t test.** Pairwise differences with confidence intervals containing zero are not significant at the 5% probability level. MAE, rf, xgbTree, nnet, bridge, LM, respectively, mean absolute error, root mean square error, coefficient of determination, random forest, extreme gradient boosting, neural network, Bayesian ridge regression, linear model.

**Table 3. Validation of the different machine learning algorithms in the testing set.**

| Metrics/ Models | LM | RF | BRR | NNET | xgbTree |
|---|---|---|---|---|---|
| $R^2$ (%) | 3.00 | 6.00 | 35.00 | 20.00 | 1.00 |
| SMAPE (%) | 1.66 | 0.21 | 0.30 | 1.64 | 0.30 |
| MAPE (%) | 4.12 | 0.25 | 0.40 | 0.90 | 0.33 |
| MAE (t ha$^{-1}$) | 42.49 | 2.28 | 3.40 | 10.06 | 3.05 |

MAE, MAPE, SMAPE, $R^2$, rf, xgbTree, nnet, bridge, LM, respectively, mean absolute error, mean absolute percentage error, symmetric mean absolute percentage error, coefficient of determination, random forest, extreme gradient boosting, neural network, Bayesian ridge regression, linear model.

followed in a decreasing order by averaged reflectance sampled on DOYs 184–189, 225–239, and 249–264 (Fig 10).

## Discussion

In this work, we used a smaller size of the training set than would be expected in a typical machine learning experiment [68, 69]. According to Gonfalonieri [58], roughly 10 times as many examples (in training set's instances) are required as there are degrees of freedom in the model in order to avoid overfitting the model to the data. Overfitting refers to an algorithm that models the training data too well, learning the detail, configuration and noise in the training data to the extent that it negatively impacts the performance of the model on new, unseen data, e.g., independent validation set. This suggests that the size of the training set used in this work would have been at least 240. However, in addition to using the "one standard error" rule of Breiman et al. [57], the models implemented in this work were multicollinearity and overfitting resistant [59], enough regression degrees of freedom were provided and multicollinearity controlled in the linear model, and several other overfitting corrective measures were anticipated. First, since our models were implemented as proof of concept, not in production, a small data set can be good enough [58]. Second, using data collected at 15-day intervals or derived from fortnightly averages reduced greatly the correlation between inputs which would otherwise make models overfitting. In addition, a Bayesian method (Bayesian ridge regression) was included in the evaluated predictive models and helped avoid overfitting; Bayesian methods are simple and hence, learn from small data sets remarkably better than more complicated models (e.g., Extreme Gradient Boosting, neural networks, and random forest) since they are essentially trying to learn less [58, 68]. The importance of Bayesian models in small samples

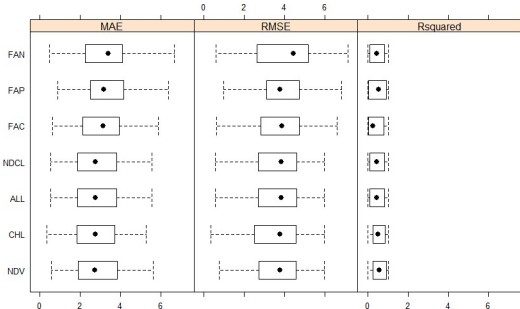

**Fig 7. Distribution of performance metrics of models based on Bayesian ridge regression and different combinations of sensor data.** MAE, RMSE, Rsquared, FAN, FAP, FAC, NDCL, CHL, NDV, ALL, respectively, mean absolute error, root mean square error, coefficient of determination, fAPAR and NDVI, fAPAR, fAPAR and CI, NDVI and CI, CI, NDVI, and all the 3 parameters (fAPAR, NDVI, CI).

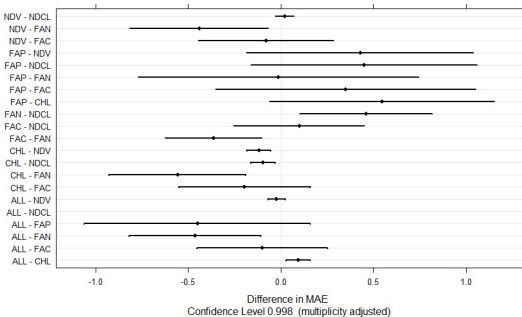

**Fig 8. Pairwise comparison of models derived from BRR as implemented in different combinations of sensor data.** FAN, FAP, FAC, NDCL, CHL, NDV, ALL, respectively, fAPAR and NDVI, fAPAR, fAPAR and CI, NDVI and CI, CI, NDVI, and all the 3 parameters (fAPAR, NDVI, CI). Pairwise differences with confidence intervals containing zero are not significant at the 1% probability level. Bonferroni correction was applied to account for the problem of multiple comparisons.

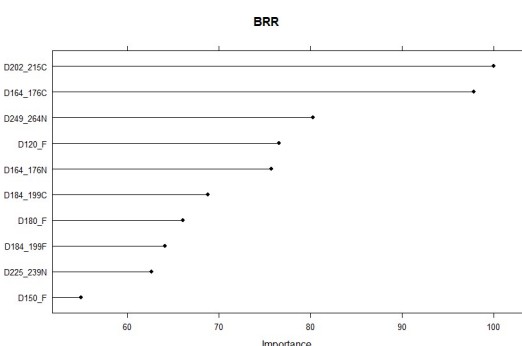

**Fig 9. Importance of features resulting from sensor data types x sampling times using Bayesian ridge regression.** BRR, D, C, F, N, respectively, Bayesian ridge regression, day of year, chlorophyll concentration index, fraction of absorbed photosynthetically active radiation, normalized difference vegetation index. Numbers or numbers linked with underscore following the letter "D" are single day of year or intervals of days of year. Refer to the text for the description of these variables.

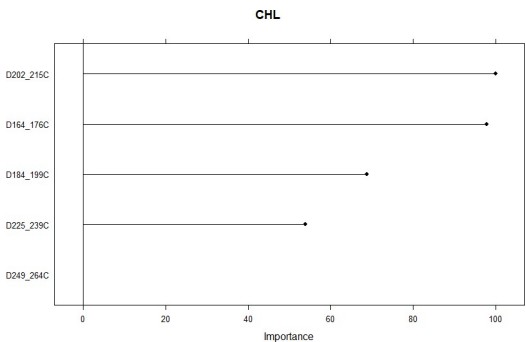

**Fig 10. Importance of features resulting from implementation of Bayesian ridge regression on CI data type across different sampling times.** CHL, D, C, respectively, chlorophyll concentration index-based Bayesian ridge regression, day of year, chlorophyll concentration index. Numbers linked with underscore following the letter "D" are intervals of days of year. Refer to the text for the description of these variables and the CHL model.

was confirmed in this work as Bayesian ridge regression performed better than more complex models such as nnet, rf, and xgbTree; Bayesian models are therefore expected to be frequently used in crop monitoring as agricultural digitalization is still at its early stages and implemented in few farmers with limited time series data. Finally, the train/test split approaches (70% *vs*. 30% training/calibration) implemented in this work is known to produce robust and unbiased performance estimates regardless of sample size [69]. In the Train/Test Split, a portion (30%) of data was separated before developing a machine learning model and then used only for validation.

Remote and proximal sensing data were used in this work to predict the in-season yields in biomass sorghum as these technologies represent the main strategies suggested for monitoring crops, like sorghum, that have a direct relation between biomass and the harvest index [30]. The modelling process implemented herein—machine learning techniques—were used to predict yields like in similar approaches currently used to develop agricultural models based on remote sensing imagery [70–72]. According to [60], machine learning techniques provide a higher accuracy and a more robust performance compared to conventional correlations as they learn to model complexity through training. In addition, remote sensing models to forecast crop yields can perform comparably or better than complex crop simulation models [4]. Different data types collected from handheld chlorophyll and NDVI meters, and from Sentinel-2 satellite constellations, were comparatively evaluated for their potential use as proxies for in-season aboveground biomass yields in sorghum. The focus of this work on yields predictions in *Sorghum bicolor* and hybrids therefrom can be justified by this species being a staple food in developing countries, a niche and a resurging crop in developed countries; in addition, due to its resilience to climate change adversities (e.g., soil moisture and nutrients scarcity), sorghum cultivation is expected to steadily increase worldwide [3]. Early within-season prediction of biomass production has positive implications including increased efficiency in the management of biomass, biofuel, and farming resources, and avoidance of commodity crises [4].

Different sorghum types were used in this study as they are similarly grown to produce aboveground commercial biomass. Combining modelling instances from different sorghum types was specifically justified by the need to mimic farming practices in the Mediterranean regions where, farmers, farming cooperatives, and other stakeholders manage these sorghum populations indiscriminately on a cropping season basis. Similar predictive analytics using instances of several different populations were reported in previous works carried out on different crop species [4, 19, 65].

The profiles of the fraction of absorbed photosynthetically active radiation, normalized difference vegetation index and chlorophyll concentration index observed in this work paralleled the evolution of leaf senescence in sorghum under the Mediterranean environment [73–76]. Leaf reflectance as a proxy for leaf chlorophyll concentration increased throughout the cropping season (Table 2) with peaks mostly in the second half of August. Since all sorghum trials reported in this study were established under a rainfed regime, and considering that the Mediterranean environments are characterized by a semi-arid climate in which summer crops rely on winter soil-stored moisture and experience post-anthesis drought stress, it can be inferred that the reflectance peaks observed towards mid-summer (second half of July) and the end of summer (second half of August) reflect the drought stress resistance and/or tolerance probably conferred by the stay-green and other resilience traits (e.g., osmotic adjustment, long and deep root architecture, etc.) expressed in these sorghum populations and witnessed in a number of other studies [73–76].

A good level of model prediction performance was obtained during cross-validation in this work, with mean MAE $\leq$ 3.00 t ha$^{-1}$ in the best models (e.g., BRR, RF, and CI-based BRR) and

$R^2$ range of 40–60% (Figs 5 and 7). During model validation (Table 3), medium values of reliability ($R^2$ = 35%) and low prediction error (e.g., MAPE = 0.40%) were obtained in the Bayesian ridge regression model. The observed medium accuracy meant that the relationship between the predicted and actual yields was not highly linear [23]; the model explained 35% of the variability that existed in the sorghum biomass yield data, while the remaining variance can be related to other environmental factors non accounted for in this study but that could give rise to differing yield responses across the farms [4]. The modeling performance metrics achieved in this work are nonetheless comparable or better than previous findings. For instance, Shafian et al. [46] came across similar accuracy in their work on sorghum yields. On the other hand, the accuracy realized in this work was greater or equal to the values reported by Gao et al. [65] in their study assessing the variability of corn and soybean yields in central Iowa using high spatiotemporal resolution multi-satellite imagery, Kayad et al. [60] in their study monitoring within-field variability of corn yield using Sentinel-2 and machine learning techniques, and by Shwalbert et al. [77] in a recent study performed in Brazil and the USA to investigate different vegetation indices derived from Sentinel-2 images to predict corn grain yield at a field scale. The latter study showed that Normalized Difference Red Edge, Green Normalized Difference Vegetation Index, and Normalized Difference Vegetation Index presented high performance to forecast field variability and provided an $R^2$ value of 32% for the universal corn yield estimation equation. In their work on modelling yields in winter wheat, sorghum and corn, using Advanced Very High Resolution Radiometer satellite (AVHRR)-derived NDVI data as proxies, Kogan et al. [20] obtained modelling biases in the order of 3%, 6%, and 8%, respectively, for corn, sorghum and wheat, which are higher than MAPE values obtained in this work.

The inference on MAE pairwise differences between the evaluated algorithms showed that the simple linear model performed poorly. Random forest showed a low bias comparable to Extreme Gradient Boosting and Bayesian Ridge Regression, but Bayesian ridge regression combined good MAE and highest $R^2$ and was therefore considered as the best machine learning algorithm (Fig 6). A similar performance of a Bayesian machine learning was reported in Habyarimana et al. [5]. The good performance of Random Forest, xgbTree, and BRR is in agreement with the findings in [4, 55, 60]. Clearly, nonparametric methods showed lower performance in this work compared to Kayad et al. [60] and Habyarimana et al. [4], and the reason can be the intrinsic properties of these algorithms. Parametric models are simpler, need less data for training, and perform satisfactorily as far as they are not deployed on complex problems. On the other hand, nonparametric models need bigger-sized (*vs.* the medium dataset used in this work) training dataset, and can be compromised by their trend to overfitting [78].

One of the main objectives of this work was to determine the comparative performance of machine learning models based on different sensor data types. Using the best identified machine learning method (Bayesian ridge regression), the model based on Sentinel-2-derived fAPAR was comparable with those based on handheld FieldScout-derived NDVI and CI, and combining sensor data types did not result in the improvement of model performance (Fig 8). Since the CI-based model outperformed NDVI-based model, it can be inferred that either Sentinel-2 fAPAR or FieldScout CI can be recommended for use as proxies of sorghum biomass yields, and for implementation in monitoring this crop as the two sensor data types performed comparably well but present distinctive use cases. Crop yield monitoring and forecasting using satellite data can be motivated by near-real-time and wide coverage, and the possibility to produce vegetation indicators at low cost, but requires expertise to access and process the original information. On the other hand, the hand-held optical chlorophyll meter is more affordable, more farmer friendly as it is easy to operate, requires little technical skills and facilities, and,

unlike the complex spectroradiometers, the handheld optical chlorophyll meter is not dependent on resource consuming calibration cycles [42]. The possibility of having alternative (Hand-held meter *vs*. Satellite imagery) crop yield forecasting solutions is important in the Mediterranean region where there is a coexistence of small and medium to big farms, individual farmers and farming cooperatives with no or few infrastructures to process satellite imageries. While it can be necessary for the extension services to use satellite imageries to monitor crops as they are expected to deal with several farms across the national territory, it can nonetheless be handy for farming cooperatives or third-party service provider to use a simple but accurate hand-held sensor to monitor their fields, at a relatively smaller scale, for a few favorable points in time during the cropping season. The superiority of model based on CI over NDVI was not expected given that the latter is the more frequently used in reflectance analyses [4]. The performance advantage of CI over NDVI can be attributed to the fact that the chlorophyll meter used in this work sampled in the red-edge while the NDVI meter sampled in the red. Red-edge spectral reflectance is associated with higher sampling resolution as it is relatively easily affected (strongly controlled) by chlorophyll concentration change compared to the red spectral reflectance [39]. In addition, NDVI data distributions show saturation towards higher reading values [47], and this can decrease the model performance.

The second half of June (DOY 164–176) and the second half of July (DOY 202–215) were the best vegetation indices sampling times in this work (Figs 9 and 10). This findings are partly in agreement with [4] in the sense that the best fortnights intervals (DOYs 164–176 and 202–215) identified in this study contain DOYs 165 and 210 that were, respectively, the second most and the least important features (regressors) reported by [4]. The observed discrepancies between the two works can be explained by the use of different data types: fAPAR (single and averaged fortnights), NDVI, and CI were used in this work, while only single DOYs fAPAR were sampled in the previous work.

## Conclusions

Crop monitoring and yield prediction can improve plant breeding, farm management and help countries avoid food and commodity crises. Modern remote and proximal sensing technologies showed a good potential to revolutionize crop monitoring, but they present the end users with different levels of complexity and/or constraints. In this work, the model based on simple easy-to-use handheld optical reflectance meter sampling chlorophyll concentration index in the red-edge at 700 nm outperformed NDVI meter-based model and was comparable to model based on Sentinel-2-derived fAPAR, while the combination of sensor data types did not bring about prediction improvement. The handheld chlorophyll meter was therefore a good alternative to Sentinel-2 satellites-derived fAPAR for the purpose of yields prediction in biomass sorghum. The month of July and the second half of June were the best times for vegetation indices sampling and biomass yield prediction. The higher reliability observed in the Bayesian ridge regression which is a parametric model, testified to the importance of the implementation of the Bayesian methods in small-sized samples. As cases of small sample size are expected to be common in the low technological industry such as agriculture [79], Bayesian models will be the methods of choice. The results achieved in this work will serve several purposes including crop genetic improvement, farm management, yields forecast, commodity planning, and extension policy refinements. The improved performance of yields prediction based on chlorophyll concentration measurements will be even better than field surveys and censuses that were conventionally used in crop monitoring but were costly and unreliable.

## Supporting information

**S1 Table. Features used in the machine learning models.** D, C, F, N, respectively, day of year, chlorophyll concentration index, fraction of absorbed photosynthetically active radiation, normalized difference vegetation index. Numbers or numbers linked with underscore following the letter "D" are single day of year or intervals of days of year. Refer to the text for the description of these variables.
(XLSX)

## Author Contributions

**Conceptualization:** Ephrem Habyarimana.

**Data curation:** Ephrem Habyarimana.

**Formal analysis:** Ephrem Habyarimana, Faheem S. Baloch.

**Funding acquisition:** Ephrem Habyarimana.

**Investigation:** Ephrem Habyarimana.

**Methodology:** Ephrem Habyarimana.

**Project administration:** Ephrem Habyarimana.

**Resources:** Ephrem Habyarimana.

**Software:** Ephrem Habyarimana.

**Supervision:** Ephrem Habyarimana.

**Validation:** Ephrem Habyarimana.

**Visualization:** Ephrem Habyarimana, Faheem S. Baloch.

**Writing – original draft:** Ephrem Habyarimana.

**Writing – review & editing:** Ephrem Habyarimana, Faheem S. Baloch.

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
