## [Decision Letter · Decision Letter 0]

15 Dec 2020

PONE-D-20-28051

Machine Learning Models Based on Remote and Proximal Sensing as Potential Methods for In-Season Biomass Yields Prediction in Commercial Sorghum Fields

PLOS ONE

Dear Dr. Habyarimana,

Thank you for submitting your manuscript to PLOS ONE. After careful consideration, we feel that it has merit but does not fully meet PLOS ONE’s publication criteria as it currently stands. Therefore, we invite you to submit a revised version of the manuscript that addresses the points raised during the review process.

We look forward to receiving your revised manuscript.

Kind regards,

Jeff Atkins

Academic Editor

PLOS ONE

Journal Requirements:

2. In your Methods section, please provide additional location information of the study sites, including geographic coordinates for the data set if available.

4. Thank you for stating the following after the Conclusion Section of your manuscript:

'Funding: This research was funded by the European Union, grant number 732064 (H2020-ICT-2016-1-innovation action and the APC was funded by the European Union through the project Data-driven Bioeconomy (www.databio.eu) and the Ministero delle Politiche Agricole, Alimentari, Forestali e del Turismo (Rome, Italy) through the project Risorse GeneticheVegetali (RGV/FAO) 2014–2019.'

'E.H. received funds from the European Union, grant number 732064 (H2020-ICT-2016-1-innovation action. The funders had no role in study design, data collection and analysis, decision to publish, or preparation of the manuscript.'

5. We note that Figure 1 in your submission contains map/satellite images which may be copyrighted.

We require you to either (a) present written permission from the copyright holder to publish these figure specifically under the CC BY 4.0 license, or (b) remove the figure from your submission:

b. If you are unable to obtain permission from the original copyright holder to publish this figure under the CC BY 4.0 license or if the copyright holder’s requirements are incompatible with the CC BY 4.0 license, please either i) remove the figure or ii) supply a replacement figure that complies with the CC BY 4.0 license. Please check copyright information on all replacement figures and update the figure caption with source information. If applicable, please specify in the figure caption text when a figure is similar but not identical to the original image and is therefore for illustrative purposes only.

6. Please include captions for your Supporting Information files at the end of your manuscript, and update any in-text citations to match accordingly. Please see our Supporting Information guidelines for more information: http://journals.plos.org/plosone/s/supporting-information

Additional Editor Comments:

Thank you for your submission and my apologies for delays caused by COVID-19. I have considered the input of both reviewers, whom I respect and value greatly. R2 brings up the point of small sample size in testing, and you do address that in your discussion. R1 brings up points about your modelling efforts and how to better clarify those efforts. Addressing those and other concerns they both raise will be helpful. To R1's point on small sample size--this is concerning, but I think framing this paper more as a proof of concept (which you do at points) would be more advisable. Your introduction reads like a review. I think carefully paying attention to where you can augment the existing text to make sure readers understand this is an attempt to push current methods towards opening up new avenues of research would be helpful. Thank you.

Reviewers' comments:

Reviewer's Responses to Questions

**Comments to the Author**

1. Is the manuscript technically sound, and do the data support the conclusions?

Reviewer #1: Yes

Reviewer #2: No

2. Has the statistical analysis been performed appropriately and rigorously? 

Reviewer #1: Yes

Reviewer #2: No

3. Have the authors made all data underlying the findings in their manuscript fully available?

Reviewer #1: Yes

Reviewer #2: Yes

4. Is the manuscript presented in an intelligible fashion and written in standard English?

Reviewer #1: Yes

Reviewer #2: Yes

5. Review Comments to the Author

Reviewer #1: The manuscript compared several crop yield models with several different remotely sensed optical features in several sorghum fields in Italy. It's an interesting topic and fit to the scope of PLOS ONE. While I have several concerns before the consideration of the publication:

My main concerns are about the feature selections used in learning model:

1.1. there is no description or discussion on how these features were selected

1.2. there are 24 features/inputs in the model, while most of them are same features, like 14 features are fAPAR that captured in different time, please add more information about how did you handle the correlations between these inputs to avoid model overfitting

1.3. The description of different learning models is inadequate, more details are needed to show the differences of these learning models and why they are used in this study

1.4. Section 3.1 should be in methodology as it's the description of model inputs

Reviewer #2: This manuscript reports the result of applying different machine learning techniques for Sorghum biomass Prediction. Authors analysed different Sentinel-2 images and collected ground data from different fields using hand held sensors for chlorophyll and NDVI measurements from 2 different years. In general, the topic is current, interesting and the manuscript has adequate information for methodology and results. However, the ground observation number is not suitable for such analysis considering the different varieties, years and fields. Moreover, I have the following specific comments:

L40: This sentence does not add value. It is just defining the word forecast by using the same word. Please revise.

L41: (simplifies), I suggest to replace it with the word support.

L71: I don't think that it is possible to estimate corn yield 4 months before harvesting. Please revise this information.

L228: How did you apply the PROSAIL model to retrieve the fAPAR?

L252-261: Move this part to your introduction section.

L280: The number of observations is very low. I suggest to collect more ground observations as it is not possible to depend on these current results.

L285-306: This information is not related to your methodology and elaborating more in a well known information. I suggest reducing this part and moving it to the introduction section.

L336 Please define where to find this information.

For all figures: Please maintain a consistent style for figure legends, X and Y labels and values.

6. PLOS authors have the option to publish the peer review history of their article (what does this mean?). If published, this will include your full peer review and any attached files.

Reviewer #1: No

Reviewer #2: No

---

## [Author Response · Author response to Decision Letter 0]

23 Dec 2020

Response to reviewers

Dear Editor, 

Thank you very much for considering my paper for publication in the journal PLOS ONE. 

It’s a pleasure for myself and on behalf of the co-author, to be able use the Editor and Reviewer constructive comments and suggestions to improve this manuscript. 

I systematically went through the comments and suggestions, providing our comments, answers, and actions we took in response to Editor and Reviewers concerns. 

Below, our answers are provided in italicized font preceded with the capital letter “R” followed by a column “:”

Cordially. 

Ephrem. 

------------------- 

PONE-D-20-28051

Machine Learning Models Based on Remote and Proximal Sensing as Potential Methods for In-Season Biomass Yields Prediction in Commercial Sorghum Fields

PLOS ONE

R: Style and file naming were corrected according to PLOS ONE standard.

 2. In your Methods section, please provide additional location information of the study sites, including geographic coordinates for the data set if available. 

R: Geographic coordinates were added to Table 1 

R: On lines 146-149 in the Manuscript file a sentence was added to explain there were formal contracts that regulated the trials in private commercial fields.

 4. Thank you for stating the following after the Conclusion Section of your manuscript:

'Funding: This research was funded by the European Union, grant number 732064 (H2020-ICT-2016-1-innovation action and the APC was funded by the European Union through the project Data-driven Bioeconomy (www.databio.eu) and the Ministero delle Politiche Agricole, Alimentari, Forestali e del Turismo (Rome, Italy) through the project Risorse GeneticheVegetali (RGV/FAO) 2014–2019.'

'E.H. received funds from the European Union, grant number 732064 (H2020-ICT-2016-1-innovation action. The funders had no role in study design, data collection and analysis, decision to publish, or preparation of the manuscript.' 

R: As suggested, funding-related information was removed from the manuscript.

R: Funding statements were included in the cover letter as suggested.

 5. We note that Figure 1 in your submission contains map/satellite images which may be copyrighted.

 We require you to either (a) present written permission from the copyright holder to publish these figure specifically under the CC BY 4.0 license, or (b) remove the figure from your submission:

R: This figure was properly cited in the manuscript. It was originally published in Agronomy (Agronomy 2019, 9(4), 203; https://doi.org/10.3390/agronomy9040203) under (CC BY) license (http://creativecommons.org/licenses/by/4.0/) which permits unrestricted use. Since the original paper was authored by myself and with a license that permits unrestricted use, there is no need for a written permission to copyright owner. These are the rules for the articles published in MDPI journals: 

copyright is retained by the authors (https://www.mdpi.com/authors/rights).

R: The text was added in the figure caption of the copyrighted Fig 1.

 6. Please include captions for your Supporting Information files at the end of your manuscript, and update any in-text citations to match accordingly. Please see our Supporting Information guidelines for more information: http://journals.plos.org/plosone/s/supporting-information

 R: The captions of the S1 Table was added at the end of the manuscript as suggested.

Additional Editor Comments:

Thank you for your submission and my apologies for delays caused by COVID-19. I have considered the input of both reviewers, whom I respect and value greatly. R2 brings up the point of small sample size in testing, and you do address that in your discussion. R1 brings up points about your modelling efforts and how to better clarify those efforts. Addressing those and other concerns they both raise will be helpful. To R1's point on small sample size--this is concerning, but I think framing this paper more as a proof of concept (which you do at points) would be more advisable. Your introduction reads like a review. I think carefully paying attention to where you can augment the existing text to make sure readers understand this is an attempt to push current methods towards opening up new avenues of research would be helpful. Thank you. 

R: Small sample size: this issue was addressed in the manuscript particularly on lines 467-479 of the manuscript. 

Reviewers' comments:

Reviewer's Responses to Questions

Comments to the Author

1. Is the manuscript technically sound, and do the data support the conclusions?

Reviewer #1: Yes

Reviewer #2: No

2. Has the statistical analysis been performed appropriately and rigorously? 

Reviewer #1: Yes

Reviewer #2: No

3. Have the authors made all data underlying the findings in their manuscript fully available?

Reviewer #1: Yes

Reviewer #2: Yes

4. Is the manuscript presented in an intelligible fashion and written in standard English?

Reviewer #1: Yes

Reviewer #2: Yes

5. Review Comments to the Author

Reviewer #1: The manuscript compared several crop yield models with several different remotely sensed optical features in several sorghum fields in Italy. It's an interesting topic and fit to the scope of PLOS ONE. While I have several concerns before the consideration of the publication:

My main concerns are about the feature selections used in learning model:

1.1. there is no description or discussion on how these features were selected

R: The process of selecting the features used in the models was explained in the Manuscript particularly on lines 246-257, 259-267, and 289-300. 

1.2. there are 24 features/inputs in the model, while most of them are same features, like 14 features are fAPAR that captured in different time, please add more information about how did you handle the correlations between these inputs to avoid model overfitting

R: Collecting data at 15-day intervals and averages reduced the correlation between inputs. On the other hand, several other steps were taken to avoid model overfitting and this was explained in the manuscript e.g., (1) Our work is a proof of concept accommodating small sample size, and, among models we included low-complexity models such as Bayesian ridge regression (Manuscript lines 274-275, 468-473), (2) implementing the “one standard error rule” and using fortnightly data (Manuscript lines 467-473). 

1.3. The description of different learning models is inadequate, more details are needed to show the differences of these learning models and why they are used in this study

R: The models were comprehensively explained in the manuscript, particularly in lines 263-278, 471-480.

1.4. Section 3.1 should be in methodology as it's the description of model inputs

R: Section 3.1 “Descriptive statistics of the features used in modelling” present the analytics outcome and should therefore remain under Results’ section. 

Reviewer #2: This manuscript reports the result of applying different machine learning techniques for Sorghum biomass Prediction. Authors analysed different Sentinel-2 images and collected ground data from different fields using hand held sensors for chlorophyll and NDVI measurements from 2 different years. In general, the topic is current, interesting and the manuscript has adequate information for methodology and results. However, the ground observation number is not suitable for such analysis considering the different varieties, years and fields. Moreover, I have the following specific comments: 

R: The ground observation number can be considered suitable as this work is a proof of concept (Manuscript lines 468-469) aiming mainly at opening new research venues (lines 123-125). In addition, small-sized samples are expected to be a rule not an exception in agricultural sciences, particularly at demonstration or pilot levels requiring commercial fields (lines 592-593). 

L40: This sentence does not add value. It is just defining the word forecast by using the same word. Please revise. 

R: the sentence was edited (Manuscript lines 41-42)

L41: (simplifies), I suggest to replace it with the word support.

R: replaced as suggested (Manuscript line 42)

L71: I don't think that it is possible to estimate corn yield 4 months before harvesting. Please revise this information. 

R: The Reviewer was right. There were typos in previous version, we fixed the errors as shown on lines 71-74 of the manuscript.

L228: How did you apply the PROSAIL model to retrieve the fAPAR?

R: We edited the text for a better reading, and the use of PROSAIL model was better described in the manuscript on lines 228-245. 

L252-261: Move this part to your introduction section. 

R: since this are short specific descriptions of the algorithms used in the work, we consider they should remain under Materials and methods”. 

L280: The number of observations is very low. I suggest to collect more ground observations as it is not possible to depend on these current results. 

R: The ground observation number can be considered suitable as this work is a proof of concept (Manuscript lines 468-469) aiming mainly at opening new research venues (lines 123-125). In addition, small-sized samples are expected to be a rule not an exception in agricultural sciences, particularly at demonstration or pilot levels requiring commercial fields (lines 592-593).

L285-306: This information is not related to your methodology and elaborating more in a well known information. I suggest reducing this part and moving it to the introduction section. 

R: since the information is about the description of the model evaluation metrics implemented in the study as can be seen in the produced tables and figures, the information should remain under Materials and methods. 

L336 Please define where to find this information. 

R: prior to arriving at the Table 2, the information in question would have been read in paragraph above. 

For all figures: Please maintain a consistent style for figure legends, X and Y labels and values. 

R: each figure is a different graphic representation and is therefore a different figure. Yet, we tried to be as consistent as we could. 

6. PLOS authors have the option to publish the peer review history of their article (what does this mean?). If published, this will include your full peer review and any attached files.

Do you want your identity to be public for this peer review? For information about this choice, including consent withdrawal, please see our Privacy Policy.

Reviewer #1: No

Reviewer #2: No

---

## [Decision Letter · Decision Letter 1]

8 Mar 2021

PONE-D-20-28051R1

Machine learning models based on remote and proximal sensing as potential methods for in-season biomass yields prediction in commercial sorghum fields

PLOS ONE

Dear Dr. Habyarimana,

Thank you for submitting your manuscript to PLOS ONE. After careful consideration, we feel that it has merit but does not fully meet PLOS ONE’s publication criteria as it currently stands. Therefore, we invite you to submit a revised version of the manuscript that addresses the points raised during the review process.

Please address the over fitting issue raised by reviewer 1, in particular how the linear model parameters are obtained.

We look forward to receiving your revised manuscript.

Kind regards,

Jie Zhang

Academic Editor

PLOS ONE

Journal Requirements:

Additional Editor Comments (if provided):

Reviewer 1 has a valid point on the fact that less number of samples the inputs can have a serious impact on the model generalisation, i.e. the over fitting problem. This is especially the case for linear model if the conventional MLR is used for finding model parameters. To cope this problem, principal component regression, or partial least square regression, or ridge regression should be used. The authors need to clarify how they obtained the linear model parameters.

Reviewers' comments:

Reviewer's Responses to Questions

**Comments to the Author**

1. If the authors have adequately addressed your comments raised in a previous round of review and you feel that this manuscript is now acceptable for publication, you may indicate that here to bypass the “Comments to the Author” section, enter your conflict of interest statement in the “Confidential to Editor” section, and submit your "Accept" recommendation.

Reviewer #1: (No Response)

Reviewer #2: All comments have been addressed

2. Is the manuscript technically sound, and do the data support the conclusions?

Reviewer #1: No

Reviewer #2: Yes

3. Has the statistical analysis been performed appropriately and rigorously? 

Reviewer #1: No

Reviewer #2: Yes

4. Have the authors made all data underlying the findings in their manuscript fully available?

Reviewer #1: No

Reviewer #2: Yes

5. Is the manuscript presented in an intelligible fashion and written in standard English?

Reviewer #1: Yes

Reviewer #2: Yes

6. Review Comments to the Author

Reviewer #1: The authors didn't address my concerns about the model overfitting issue at all. Instead of rebuttal using statement in the manuscript (which have no proof/reference, such as '15-day intervals and averages reduced the correlation between inputs'), please provide statistical evidence that your model is not overfitting, such as training accuracy and testing accuracy when you do machine learning modeling. or at least provide "adjusted R-square" which is adjusted for the number of parameters, or ANOVA test to proof the independence of your inputs, or at least correlation scatter plots between inputs. But any way, 24 parameters for 23 observations are way too many to avoid overfitting. Please read this post by Jim Frost for reference https://statisticsbyjim.com/regression/overfitting-regression-models/.

Reviewer #2: Authors replied to my comments, improved their manuscript and I have no further comments.

7. PLOS authors have the option to publish the peer review history of their article (what does this mean?). If published, this will include your full peer review and any attached files.

Reviewer #1: No

Reviewer #2: No

---

## [Author Response · Author response to Decision Letter 1]

11 Mar 2021

A copy of the Responses to Reviewers was uploaded in the system but, below are the responses as diretly entered to the attention of the Journal office.

Additional Editor Comments (if provided):

Reviewer 1 has a valid point on the fact that less number of samples the inputs can have a serious impact on the model generalisation, i.e. the over fitting problem. This is especially the case for linear model if the conventional MLR is used for finding model parameters. To cope this problem, principal component regression, or partial least square regression, or ridge regression should be used. The authors need to clarify how they obtained the linear model parameters. 

RESPONSE: We agree with the Editor on these issues of overfitting and multicollinearity, particularly in the linear model. Usually a few options are available in order to minimize the effect of multicollinearity and these can include: using models that are resilient to sizeable between-feature correlations, such as PLS, PCA, and other machine learning algorithms least squares. Alternatively, as we did in this work, we can identify and remove those features that contribute the most to the between-feature correlations. 

All models implemented in this study were multicollinearity resistant except the linear model [60] which therefore required additional measures to avoid overfitting (Lines 293-301). In the linear model, in order to minimize the effect of collinearity, solve the issue of the number of predictor variables being greater than the sample size, estimate all parameters including the constant, and finally be able to fit the overall model during the training stage, we opted to reduce the number of predictors by using an algorithm to remove a subset of those features involved with the most high pairwise correlations such that the sample (training set) size is two more than the number of predictors (allowing for a residual degree of freedom), and all of the remaining pairwise Pearson correlation coefficients are below a 0.90 threshold [60].

Reviewer #1: The authors didn't address my concerns about the model overfitting issue at all. Instead of rebuttal using statement in the manuscript (which have no proof/reference, such as '15-day intervals and averages reduced the correlation between inputs'), please provide statistical evidence that your model is not overfitting, such as training accuracy and testing accuracy when you do machine learning modeling. or at least provide "adjusted R-square" which is adjusted for the number of parameters, or ANOVA test to proof the independence of your inputs, or at least correlation scatter plots between inputs. But any way, 24 parameters for 23 observations are way too many to avoid overfitting. Please read this post by Jim Frost for reference https://statisticsbyjim.com/regression/overfitting-regression-models/. 

RESPONSE: We thank the Reviewer as his remarks helped us better rephrase and explain how we prepared the data and implemented the models (Lines 293-301). All models implemented in this study were multicollinearity resistant except the linear model [60] and measures to avoid overfitting were applied. For instance, the “one standard error” rule of Breiman et al. [61] was used, and the methods’ built-in features were invoked to automatically select features, tune hyperparameters to the data set, and select the best final model for the downstream validation step. In the process of data preparation, zero-variance features were removed and those remaining were centered and scaled in order to avoid features with zero or near-zero variance which can behave like second intercepts [63]. In the linear model, in order to minimize the effect of collinearity, solve the issue of the number of predictor variables being greater than the sample size, estimate all parameters including the constant, and finally be able to fit the overall model during the training stage, we opted to reduce the number of predictors by using an algorithm to remove a subset of those features involved with the most high pairwise correlations such that the sample (training set) size is two more than the number of predictors (allowing for a residual degree of freedom), and all of the remaining pairwise Pearson correlation coefficients are below a 0.90 threshold [60]. 

Several metrics of model performance were provided throughout the manuscript e.g., in Figures 5 and 7 for the model calibration stage, in Figures 6 and 8 and Table 3 for the validation stage.

---

## [Editor Report · Decision Letter 2]

12 Mar 2021

Machine learning models based on remote and proximal sensing as potential methods for in-season biomass yields prediction in commercial sorghum fields

PONE-D-20-28051R2

Dear Dr. Habyarimana,

We’re pleased to inform you that your manuscript has been judged scientifically suitable for publication and will be formally accepted for publication once it meets all outstanding technical requirements.

Kind regards,

Jie Zhang

Academic Editor

PLOS ONE

Additional Editor Comments (optional):

The authors have adequately addressed the reviewers and editor's comments and the revised manuscript can be accepted.
---

## [Editor Report · Acceptance letter]

16 Mar 2021

PONE-D-20-28051R2 

Machine learning models based on remote and proximal sensing as potential methods for in-season biomass yields prediction in commercial sorghum fields 

Dear Dr. Habyarimana:

I'm pleased to inform you that your manuscript has been deemed suitable for publication in PLOS ONE. Congratulations! Your manuscript is now with our production department. 

Kind regards, 

on behalf of

Dr. Jie Zhang 

Academic Editor

PLOS ONE